# A Systematic Review of the Effect of Therapeutic Drug Monitoring on Patient Health Outcomes during Treatment with Carbapenems

**DOI:** 10.3390/antibiotics11101311

**Published:** 2022-09-27

**Authors:** Timothy N. Luxton, Natalie King, Christoph Wälti, Lars J. C. Jeuken, Jonathan A. T. Sandoe

**Affiliations:** 1School of Biomedical Sciences, University of Leeds, Leeds LS2 9JT, UK; 2Leeds Institute of Health Sciences, University of Leeds, Leeds LS2 9JT, UK; 3School of Electronic and Electrical Engineering, University of Leeds, Leeds LS2 9JT, UK; 4Leiden Institute of Chemistry, Leiden University, P.O. Box 9502, 2300 RA Leiden, The Netherlands; 5School of Medicine, University of Leeds, Leeds LS2 9JT, UK

**Keywords:** carbapenem, beta-lactam, meropenem, therapeutic drug monitoring, clinical outcomes, antimicrobial resistance, personalised medicine

## Abstract

Adjusting dosing regimens based on measurements of carbapenem levels may improve carbapenem exposure in patients. This systematic review aims to describe the effect carbapenem therapeutic drug monitoring (TDM) has on health outcomes, including the emergence of antimicrobial resistance (AMR). Four databases were searched for studies that reported health outcomes following adjustment to dosing regimens, according to measurements of carbapenem concentration. Bias in the studies was assessed with risk of bias analysis tools. Study characteristics and outcomes were tabulated and a narrative synthesis was performed. In total, 2 randomised controlled trials (RCTs), 17 non-randomised studies, and 19 clinical case studies were included. Significant variation in TDM practice was seen; consequently, a meta-analysis was unsuitable. Few studies assessed impacts on AMR. No significant improvement on health outcomes and no detrimental effects of carbapenem TDM were observed. Five cohort studies showed significant associations between achieving target concentrations and clinical success, including suppression of resistance. Studies in this review showed no obvious improvement in clinical outcomes when TDM is implemented. Optimisation and standardisation of carbapenem TDM practice are needed to improve intervention success and enable study synthesis. Further suitably powered studies of standardised TDM are required to assess the impact of TMD on clinical outcomes and AMR.

## 1. Introduction

Antimicrobial resistance (AMR) has become a global concern. It has been predicted that by 2050 10 million deaths per year globally will be a result of antimicrobial resistant infections [1]. In 2019, an estimated 4.95 million deaths were associated with bacterial AMR [2]. The emergence of AMR can be exacerbated by inappropriate dosing of antibiotics [3]. Therapeutic drug monitoring (TDM) is the practice of measuring a drug and adjusting dosing regimens, based on the measured amount, to reach a target concentration [4]. Pharmacokinetic/pharmacodynamic (PK/PD) indices can help describe the relationship between the dose, concentration, and outcome [5]. Thus, where it is known that certain concentrations result in improved clinical outcomes, the PK/PD indices can be used as concentration targets to adjust doses towards. PK/PD indices also reflect different antibiotic action: time-dependent antibiotics (beta-lactams) are measured based on the amount of time unbound antibiotic is above the minimum inhibitory concentration (MIC) (ƒT_>MIC_), where a fraction of the antibiotic is reversibly bound to proteins; concentration-dependent antibiotics (aminoglycosides) are measured on the ratio of peak concentration to MIC (C_max_/MIC); and antibiotics that display time and concentration-dependent action are measured on the area under the curve to MIC (AUC/MIC) [6]. TDM is already carried out on some antibiotics such as aminoglycosides and glycopeptides [7]. Previous research has shown achieving C_max_/MIC targets [8,9,10], and AUC/MIC targets [11,12] in aminoglycoside treatment improves clinical outcomes. Extended high AUC exposures and high trough concentrations over a prolonged period have been associated with nephrotoxicity [5]. In glycopeptides, higher AUC/MIC ratios are associated with improved clinical outcomes [13,14,15]. Clinical studies of patients treated with beta-lactams have shown an association between clinical outcomes and beta-lactam exposure [16,17,18,19,20,21,22,23,24,25]. Because of the different PK/PD properties of different beta-lactams, it cannot be assumed that effectiveness of TDM of one beta-lactam will translate to all beta-lactams, so evaluations are needed by antibiotic class. Despite evidence of associations between clinical outcomes and PK/PD indices, few systematic reviews have assessed the impact antibiotic TDM has on clinical outcomes and impacts on AMR [26,27,28]. 

Carbapenems are effective antimicrobial agents against Gram-positive and Gram-negative bacteria and display a broad spectrum of activity [29]. Being highly effective, but broad-spectrum agents, carbapenems are considered a last resort antibiotic. All carbapenems are on the World Health Organization’s ‘watch’ list, that refers to broad-spectrum agents that have a higher potential for resistance [29,30,31]. Carbapenem combinations meropenem–vaborbactam and imipenem–cilastatin–relebactam are included on the ‘reserve’ list, and are kept back for multidrug-resistant infections [30,31]. 

Usually dosing regimens for carbapenems are standardised and based on the susceptibility of the most likely causative pathogen and data from pharmacokinetic and pharmacodynamic (PK/PD) studies carried out in healthy volunteers [32]. However, significant PK/PD variability, particularly in the critically ill, has been shown with carbapenems and other beta-lactams [33,34,35,36]. The combination of PK/PD variability and standardised dosing regimens can result in inappropriate prescribing, doses that are either too high or too low, which can lead to less effective treatment and the emergence of AMR. Optimal stewardship of these agents is essential in the fight against AMR, particularly in last-line antibiotics, such as carbapenems. Clinical studies have reported associations between PK/PD indices and clinical outcomes in carbapenems [16,17,19,24,25]; however, carbapenem TDM is a complex intervention and there are a number of factors that can impact its effectiveness [26,37]. This systematic review assesses carbapenem TDM practice, clinical outcomes, and the impact on the emergence of AMR. 

## 2. Results

### 2.1. Search Results

The search strategy identified 1086 studies and the full texts of 355 were screened for eligibility in the review; 38 were included overall (see Figure 1). After de-duplication, no additional studies were found from the hand-search of relevant reviews. Of the 38 included studies, 2 were RCTs, 17 were non-randomised observational studies, and 19 were clinical case studies. Some studies that measured carbapenem concentration and reported clinical outcomes initially appeared to meet criteria but were excluded as dosing regimens were not influenced by the measurement result.

### 2.2. Randomised Controlled Trials 

#### 2.2.1. Quality Assessment

The risk of bias of two RCTs was assessed using the RoB2 assessment tool [38], results shown in Table 1. Both scored an overall ‘High’ risk of bias. As clinical outcomes were not the primary outcomes of either study, these studies were not powered sufficiently to identify differences in clinical outcomes [39,40]. 

#### 2.2.2. Study Characteristics and Results

The characteristics of the included RCTs are summarised in Appendix A. Patient populations included: burns patients [39], and patients with normal kidney function [40]. There was little difference in clinical outcomes seen in patients where carbapenem TDM was performed, compared to patients who received standard care. However, both RCTs were not powered to identify clinical outcomes. Additionally, important aspects of TDM in both RCTs were not comparable: PK/PD targets, dose-adjustment protocols, and TDM sampling protocols were all different. 

**Figure 1 antibiotics-11-01311-f001:**
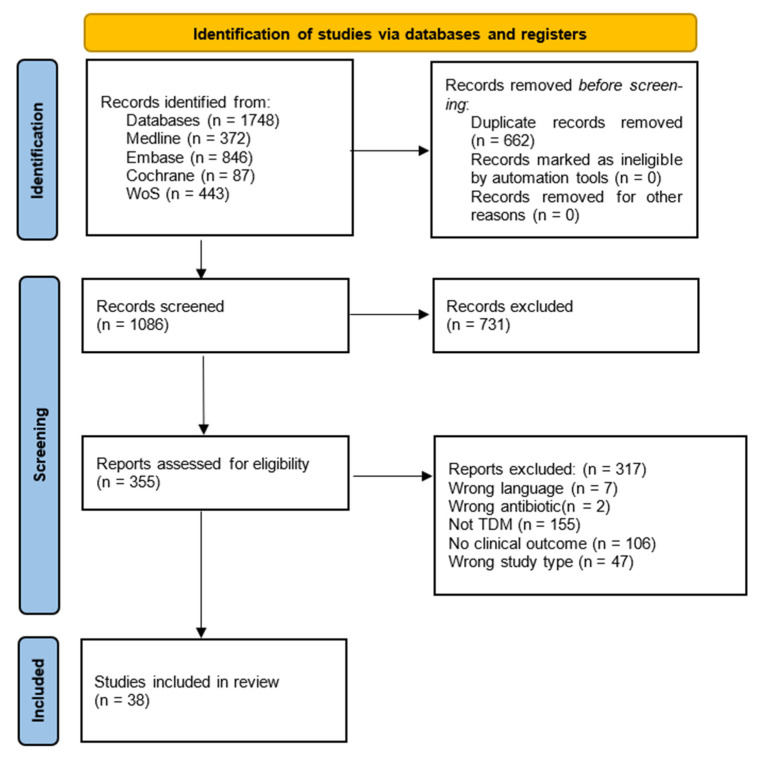
Literature search process in accordance to the Preferred Reporting Items for Systematic Reviews and Meta-Analyses (PRISMA) guidelines [41].

The RCT performed by Fournier et al. investigated the impact of using TDM in beta-lactam treatment in burns patients and TDM improved target attainment compared to the standard care group [39]. By comparing the initial carbapenem concentration to subsequent results, they showed a 12% improvement in target attainment in the TDM intervention group, compared to a 3% improvement in the standard care group. This improvement in target attainment, however, did not correspond to an improvement in clinical outcomes. The proportion of infections cured were largely the same across both groups in infections treated with meropenem, imipenem-cilastatin, and ertapenem. This study was limited by an unexpected decrease in the number of admitted patients, and it ended up underpowered to detect significant pharmacological or clinical differences between the arms; a sample size of 90 patients was required for a suitably powered study, only 38 patients were eventually recruited. Additionally, there was no definition of clinical cure, this introduced a bias in the measurement of the outcome as different clinicians may have assessed clinical resolutions differently. 

De Waele et al. carried out an RCT looking at the differences meropenem and piperacillin TDM made in patients with normal renal function [40]. Antibiotic concentrations were adjusted in the intervention group to meet a target of 100% ƒT>4xMIC. TDM improved target attainment compared to standard care. Initially the intervention group and the standard care group had a target attainment of 9.5% and 20%, respectively. After 72 h, 58% of the intervention group met the target compared to 16% in the standard care group, [*p* = 0.007]. A trend towards reduced bacterial pathogen persistence was observed (1 patient in the TDM intervention group versus 5 in the standard care group [*p* = 0.093]). Improvement in median Sequential Organ Failure Assessment (SOFA) score was larger in the intervention group, while Intensive Care unit (ICU) and 28-day mortality were the same across both groups (Intervention: ICU mortality, 1/6; 28-day mortality, 1/6. Control: ICU mortality: 1/7; 28-day mortality, 1/7, mortality data provided by the author). As with the previous RCT, the number of patients was small (*n* = 13, receiving meropenem), the study was not powered to identify differences in clinical outcomes and the reported clinical outcomes in the TDM group were not significantly different.

### 2.3. Non-Randomised Studies 

#### 2.3.1. Quality Assessment

Risk of bias according to the ROBINS I tool in the 17 non-randomised studies is presented in Table 2. All of the studies scored a serious or critical risk of bias.

#### 2.3.2. Study Characteristics and Outcomes

The characteristics of the included non-randomised studies are summarised in Appendix A. Across the included studies there were a number of different populations studied: paediatric patients [43,49], critically ill patients [47,52,53,55,56,57,58,59], burns patients [44,46,51], patients administered antibiotics for a suspected or confirmed infection [54], oncohaematological patients [50], patients receiving Continuous Renal Replacement Therapy (CRRT) [45], and patients with a carbapenemase-producing *K. pneumoniae* infection [48]. No clear benefit to clinical outcomes from the use of carbapenem TDM was identified from these studies.

TDM sampling protocols varied across studies, the majority of studies did not have a protocol for the frequency of TDM measurement and did not specify how frequently samples were taken [44,45,46,47,49,51,54,56,57,58]. Some studies reported that some patients received multiple TDM measurements, however multiple sampling was only carried out in 47.4%, [55] 31% [52], and 21.6% [53] of patients. A few studies reported median numbers of TDM measurements [48,50], or a minimum number of samples per patient [43]. One paper excluded patients with less than two TDM measurements [59].

All studies measured carbapenem blood levels, in plasma or serum, by liquid chromatography-based methods, either coupled to ultraviolet spectroscopy or tandem mass spectrometry. There was considerable variation in pharmacological targets: of the 17 observational studies, 11 different pharmacological targets were used, shown in Table 3 (also includes targets used in RCTs).

Most studies intended to determine pathogen susceptibility through susceptibility testing but when this was not available, i.e., when a pathogen could not be isolated, then ‘surrogate’ MICs were used. These were predominately based on European Committee on Antimicrobial Susceptibility Testing (EUCAST) clinical breakpoint data. 

In most studies, protocols to adjust dosing were not specified [44,45,46,48,49,50,56,57,58], or dose adjustments were at clinician’s discretion guided by antibiotic concentration measurements [54,55,59]. In other studies, a general protocol was described, stating that dosing was accordingly increased or decreased [43,51], or a more specific protocol was used, where dosing changes were pre-specified by a certain percentage increase, for example [47,52,53]. 

Few studies showed that TDM improved target attainment, i.e., by showing that target attainment increased after adjusting dosing regimens. Improvement in target attainment was only seen in two studies [48,51], variable success was seen in four studies [46,49,53,57], and no difference was seen in one study [52]. Differences in target attainment after dose adjustment was not reported in the remaining studies [43,44,45,47,50,55,58,59]. 

Of the studies that included a comparator group, evidence of clinical improvement by TDM was variable (Table 4). One study saw clinical improvements in markers for infection and hospital length of stay in the TDM group [58]. One study saw no difference in clinical outcomes between groups [44], and one study saw significantly longer length of stay in the TDM group [46]. 

Two studies investigated the effect of carbapenem TDM on AMR. Suppression of resistance was associated with achieving target concentrations (*f*T>4xMIC [57], and C_SS_/MIC ≥ 5) [56].

### 2.4. Clinical Case Studies

#### 2.4.1. Quality Assessment

The risk of bias of 19 clinical case studies was assessed using the OHAT Risk of Bias Rating Tool for Human and Animal Studies [60]. Due to the nature of clinical case studies, where there are a small number of selected patients with no ability to analyse influences from confounding factors and no control comparisons, there was a high risk of bias. All the studies scored ‘Definitely High’ for their associated risk of bias.

#### 2.4.2. Study Characteristics and Outcomes

The included studies are summarised in Appendix A [61,62,63,64,65,66,67,68,69,70,71,72,73,74,75,76,77,78,79]. Best practice for carbapenem TDM is not known, which has led to differences in the way it is carried out, particularly concerning pharmacological targets, dose-adjustment protocols, and frequency of antibiotic concentration measurement. These studies show, by measuring carbapenem concentrations, a dose adjustment is often needed. Out of the 27 patients described in the studies, dose adjustments were required in 21 (78%) due to inadequate antibiotic levels, antibiotic levels were shown to be adequate in 5 patients (19%), and in 1 patient it was unclear whether doses were adjusted or not. Of those that needed a dose adjustment and where the pathogen MIC was determined, levels met the target in 12/16 (75%) subsequent TDM measurements, and 4 did not meet the target. A positive clinical outcome was seen in 22/26 patients (85%) and treatment was stopped for 1 patient due to COVID-19. Of the patients where a positive clinical outcome was not reported, three did not reach target antibiotic levels, even after dose adjustments. 

Emergence of antibiotic resistance was reported in 3/27 (11%), and a decrease in susceptibility was reported in an additional 2 cases.

## 3. Discussion

The currently published literature provides no strong evidence to conclude that carbapenem TDM results in improved patient health outcomes, nor reduces the emergence of AMR. However, most studies were primarily powered to identify pharmacological impacts of carbapenem TDM and underpowered to detect improved health outcomes. In spite of this, there were some suggestions that carbapenem TDM could improve health outcomes: Aldaz et al., included the largest number of patients and a more robust confounding factor analysis than other non-randomised studies [58], and found TDM was associated with significant improvements in normalisation of markers for infection, such as C-reactive protein levels, and procalcitonin levels, as well as significantly shorter length of stays [58]. Two observational studies assessed the impact of TDM on AMR, both of which found statistically significant associations between AMR emergence and carbapenem concentrations that met the pharmacological target [56,57].

Due to the wide variability in TDM practice, as well as the high risk of bias, a meta-analysis was not possible. A meta-analysis was performed in Lechtig-Wasserman et al.’s systematic review of carbapenem TDM use in critically ill patients, which also found that there was not strong evidence to show clinical impacts of carbapenem TDM [27]. However, the appropriateness of this approach can be questioned because of the same wide variation in practice and risk of bias in their included studies: one study was a conference abstract with limited information on their TDM practices [80], and the others, included in the meta-analysis, carried out TDM differently [47,58]. Aside from the different pharmacological targets (100% ƒT>MIC [47] vs 100% ƒT>4-5xMIC [58]), there were critical risks of bias in confounding [47] and selection [47], as well as imbalances in patient exclusions between intervention arms, the impact of which was not assessed [58]. 

Previous research has shown that beta-lactam TDM improves target attainment [33], and previous clinical studies have shown associations between outcomes and PK/PD indices [16,17,19,24,25]. However, this review has identified that, whilst there is a previously observed association between outcomes and carbapenem target concentrations, there are no obvious improvements in outcomes when carbapenem TDM is implemented. This may be a result of study design, with inadequate power to detect a difference and/or how carbapenem TDM is carried out. To ensure that concentration targets are being met over the course of the treatment follow-up measurements must be taken. Just three of the included studies demonstrated improvements in target attainment [40,48,51], and frequency of concentration measurement was commonly not reported. A hurdle to successful TDM may be the accessibility of TDM services. The current gold-standard is HPLC-MS/MS, this requires specialised equipment, trained personnel, and time to prepare samples. More accessible TDM services, using a lab-based test that utilises commonly available equipment, such as an ELISA, or a ward-based point of care test, may allow for more frequent concentration measurements; to ensure concentrations are within target ranges over the course of treatment. This would be especially pertinent in patients with dynamic renal functions, or in patients receiving RRT [16].

This highlights the importance of the TDM clinical pathway, the steps required to carry out TDM: sampling, measurement, analysis of result, dosing regimen adjustment, follow-up measurements. Previous to this systematic review, TDM practice and the lack of standardisation in carbapenem TDM has not been discussed. In the included papers, timing of the first antibiotic concentration measurement differed, sampling frequency was rarely reported, and it differed in those studies that did report it. Additionally, the pharmacological targets, as well as definitions for ‘MIC’ for targets were inconsistent. This can result in large differences in target carbapenem concentrations, particularly if a measured MIC value is much lower than a non-species-specific clinical breakpoint. Dose-adjustment protocols required for carbapenem levels to reach targets were seldom reported and there was very little evidence reported of the effectiveness of dose adjustments. When carrying out TDM in patients, a number of concentration measurements are required throughout the treatment. This is to ensure that concentrations remain in the desired range, or to ensure dose adjustments are effective, which is particularly pertinent in patients with dynamic renal function.

TDM is a complex intervention and each step of TDM practice, from sampling to response, can have an impact on the success of the intervention. If sampling is infrequent, then there is less opportunity to respond to changing patient physiology, which is particularly important in critically ill patients and those with changing renal function. If the dose adjustment is not effective at reaching the target concentration, or if there are no data collected to show that TDM results subsequent to dose adjustments are within target ranges, then there will be little difference to standard care. If a MIC surrogate is used in place of a measured susceptibility, then a patient might receive a much larger dose than needed which will increase toxicity, compounded with a ƒT>4-10xMIC target. Differences between studies in the aforementioned variables of TDM will result in a confounded interventional outcome.

What may have diluted the ability of this review to see a clear impact of carbapenem TDM is in studies that included carbapenem TDM to treat organisms that are typically resistant to carbapenems, such as: MRSA [50,74], MRSE [50], and *Staphylococcus epidermidis* [73]. It is likely that certain conditions will be best suited for carbapenem TDM, such as to treat an isolated pathogen where the susceptibility is known as a monotherapy. To provide better evidence of when to implement carbapenem TDM, comparative studies are needed to identify whether carbapenem TDM is more effective in ‘ideal’ conditions (for example, in patients with bloodstream infection caused by Enterobacteriales), compared to conditions where pathogen identity and susceptibility are not known. Patients with abnormal renal function may be an ‘ideal’ patient group to benefit from carbapenem TDM. Carbapenem concentrations have previously been shown to be affected by renal function [81,82]. Renal failure and augmented renal function may result in a more unpredictable antibiotic elimination profile. Despite inconsistencies in TDM practice and results, studies including patients with impaired renal function showed: a need of dose adjustments in renally impaired [53]; impairment was associated with excessive amounts of antibiotic exposure [52,59], and increased mortality [54,59]. Studies that included patients with Augmented Renal Clearance (ARC) (Glomerular Filtration Rate (GFR) >130 mL/min/1.73 m^2^) showed: patients required a higher than licensed dose and more antibiotic [47]; a need for dose adjustments [53]; ARC was associated with failure to reach PK/PD targets [52], increased mortality [50], and the development of AMR [56]. RCTs are needed to assess whether TDM can improve outcomes in patients with abnormal renal function.

This review highlights challenges in the procedural and service delivery elements of TDM: the importance of timely initial antibiotic concentration measurement, and follow-up sampling, the need for standardised protocols for carbapenem TDM, frequency of sampling, susceptibility testing, dose adjustments, renal function considerations, and target concentrations; all of these elements can have a direct impact on the effectiveness of carbapenem TDM. Further studies are needed to support the, that timely carbapenem TDM can help suppress AMR and improve patient outcomes.

### Limitations

There are several limitations of this systematic review: (1) A proportion of the included studies reported combined outcomes for carbapenem antibiotics and non-carbapenem antibiotics, mostly other beta-lactams. (2) Both RCTs, and numerous observational studies were not powered to identify clinical differences. This resulted in confounding biases in the observational studies. (3) Due to the wide variation in how TDM was carried out, it was not possible to conduct a meta-analysis on these studies. (4) Inclusion of studies where carbapenem antibiotics were used to treat pathogens that are inherently resistant to carbapenems.

## 4. Materials and Methods

This systematic review was reported following Preferred Reporting Items for Systematic Reviews and Meta-Analyses (PRISMA) guidance [41]. The protocol was registered on PROSPERO (ID: CRD42020202800).

Studies investigating carbapenems and TDM were identified from the MEDLINE, EMBASE, Cochrane, and Web of Science databases. The search was conducted from database inception to 07 Feb. 2022. The search terms and strategy are shown in Appendix A. A hand search was also carried out of references of relevant reviews.

### 4.1. Definitions

Therapeutic drug monitoring (TDM) was defined as the measurement of an antibiotic used by healthcare professionals to alter the administration of the drug (dose, frequency, or route) [4].

### 4.2. Inclusion and Exclusion Criteria

Population: Patients treated with carbapenem antibiotics. Condition: Suspected or confirmed infection, as indicated by carbapenem treatment. Intervention: TDM practice to modify the dosing regimen of prescribed carbapenems. Comparator/Control: Patients treated using standard care, where dosing regimens were not influence by TDM results. Type of study: Randomised controlled trials (RCTs), non-randomised trials, cohort studies, quasi-experimental studies, retrospective, observational studies, and clinical cases were all eligible for inclusion. Studies were excluded if not written in the English language; studies not related to carbapenems; or studies not reporting clinical outcomes. Animal studies, conference abstracts and reviews were also excluded.

### 4.3. Selection and Data Extraction

EndNote X9 software (Clarivate Analytics) was used to deduplicate and manage all references. The references were screened by one reviewer (TL), 10% of the references were randomly selected using a random number generator, for screening by a second reviewer (JS) to ensure consistency. Any discrepancies between the selections were resolved through discussion with the wider team. The full-text screen was carried out by a single reviewer (TL) with 10% of the studies screened by the second reviewer (JS).

### 4.4. Data Collection

Data were collected by hand by reviewer TL and validated by the second reviewer (JS). Collected intervention data consisted of: antibiotic, administration method, dose and dose frequency, duration of therapy, antibiotic quantification method, bodily fluid used for antibiotic quantification (e.g., saliva, plasma, interstitial fluid), the fraction quantified—the free or total fraction, frequency of quantification, dose-adjustment protocol, planned antibiotic target level, and target attainment. Co-intervention data were also collected for concomitant antibiotics, and renal replacement therapy. If co-intervention data were missing, it was assumed that it was not present in the therapy. Study data collected consisted of: study type, population, population size, intervention group size, control group size, microbiological confirmation of infection, pathogen(s), study location (in hospital or community based study). Outcome data collected consisted of: mortality, in-hospital stay, length of stay on ICU, acute kidney injury, toxicity or other adverse effects, treatment effectiveness—that is the resolution of signs or symptoms of infection, duration of treatment, readmission, target attainment, emergence of antibiotic resistance. If outcome data were missing, it was assumed that the outcome measure was not recorded. Missing summary data or missing outcome data were addressed in the quality assessment.

### 4.5. Quality Assessment

The risk of bias analysis was carried out independently by reviewers TL and JS using the following tools: for RCTs, the Revised Cochrane Risk-of-Bias Tool for Randomised Trials (RoB2) assessment tool [38]; for non-randomised intervention studies, the Risk of Bias in Non-Randomised Studies (ROBINS-1) assessment tool [42]; for case studies, the Office of Health Assessment and Translation (OHAT) assessment tool [60]. Bias was assessed based on the clinical outcomes reported, irrespective if these outcomes were reported as primary aims or secondary aims.

### 4.6. Data Analysis

The studies were allocated into three categories based on the level of evidence, RCTs, non-randomised cohort studies, and case studies. To minimise impact of confounding factors on RCT results, studies were grouped by study design [83]. Study characteristics were tabulated. The following intervention characteristics were used to compare studies and assess synthesis suitability: pharmacological targets, dose-adjustment protocols, sampling initiation and frequency, type of administration, co-interventions, and susceptibility definitions. Health outcomes of studies that included a comparison group were tabulated. For studies scoring a low or moderate risk of bias, a meta-analysis would be carried out where pharmacological targets and dose-adjustment protocols were consistent.

## 5. Conclusions

Despite previous research demonstrating improvements in pharmacological target attainment from carbapenem TDM and observed associations between achieving target concentrations and clinical outcomes, this systematic review found no strong evidence of a beneficial effect of carbapenem TDM on health outcomes, including AMR. Whilst previous reviews have reported the inconclusiveness of the evidence of carbapenem TDM on clinical outcomes [27], this is the first systematic review to comprehensively analyse carbapenem TDM methodology and highlight the inconsistencies in practice, which may be the reason, alongside inadequately powered studies, why expected clinical improvements from TDM have not been seen in studies. Further studies are needed to determine best practice, such as: when to start concentration measurements, how frequently to measure concentrations, and how best to adjust doses. Adequately powered RCTs of the clinical utility of carbapenem TDM are needed, and to investigate impacts on AMR. A RCT protocol to examine clinical outcomes of beta-lactam TDM, including meropenem, has already been published that may provide more evidence of the impact of carbapenem TDM on health outcomes [84].

This review is the first to discuss the potential importance of TDM practice for clinical success, as well as assess the impact carbapenem TDM has on AMR. Where a gap in the literature has been highlighted, with only two studies assessing AMR impacts. It is known that inappropriate dosing can drive AMR, TDM is a tool to optimise dosing; therefore, it stands to reason that TDM can be a tool to suppress AMR, further studies are needed to investigate this hypothesis.

## 6. Patents

This section is not mandatory but may be added if there are patents resulting from the work reported in this manuscript.

## Figures and Tables

**Table 1 antibiotics-11-01311-t001:** Revised Cochrane Risk-of-Bias tool for Randomised Controlled Trials (RoB2) [38].

Study	Bias in Randomisation Process	Bias in Deviations from Intended Interventions (Effect of Assignment to Intervention)	Bias in Deviations from Intended Interventions (Effect If Adhering to Intervention)	Bias Due to Missing Outcome Data	Bias Due to Measurement of Outcome	Bias in Selection of Reported Result	Overall Result
Fournier et al. 2018 [39]	Some Concerns	Low	High	Low	High	Some Concerns	High
De Waele et al. 2014 [40]	Some Concerns	Low	High	Low	Low	Some Concerns	High

**Table 2 antibiotics-11-01311-t002:** Quality Assessment of non-randomised studies as per the Risk of Bias in Non-Randomised Studies of Interventions (ROBINS-I) assessment tool [42].

Study	Bias Due to Confounding	Bias in Selection of Participants	Bias in Classification of Interventions	Bias Due to Deviations from Intended Interventions	Bias Due to Missing Data	Bias in Measurement of Outcomes	Bias in Selection of the Reported Result	Overall Result
Cies et al. 2018 [43].	Critical	Low	NA	Serious	Low	Low	Moderate	Critical
Machado et al. 2017 [44].	Serious	Low	Low	NI	Low	Low	Moderate	Serious
Economou et al. 2017 [45].	Critical	Low	NA	Serious	Low	Low	Moderate	Critical
Fournier et al. 2015 [46].	Critical	Low	Serious	Critical	Low	Low	Moderate	Critical
McDonald et al. 2016 [47].	Critical	Critical	NA	NI	Low	Serious	Moderate	Critical
Pea et al. 2017 [48].	Serious	Low	NA	Moderate	Low	Low	Moderate	Serious
Cojutti et al. 2015 [49].	Critical	Low	NA	Serious	Low	Low	Moderate	Critical
Cojutti et al. 2018 [50].	Serious	Low	NA	Serious	Low	Low	Moderate	Serious
Patel et al. 2012 [51].	Critical	Low	NA	NI	Low	Serious	Moderate	Critical
Wong et al. 2018 [52].	Critical	Low	NA	Critical	Moderate	Serious	Moderate	Critical
Roberts et al. 2010 [53].	Critical	Low	NA	Serious	Low	Low	Moderate	Critical
Bricheux et al. 2019 [54].	Serious	Low	NA	Serious	Low	Low	Moderate	Serious
Schoenenberger-Arnaiz et al. 2019 [55].	Critical	Low	NA	Critical	Low	Low	Moderate	Critical
Gatti et al. 2021 [56].	Serious	Critical	NA	Low	Low	Low	Moderate	Critical
Al-Shaer et al. 2020 [57].	Serious	Critical	NA	Critical	Low	Low	Moderate	Critical
Aldaz et al. 2021 [58].	Moderate	Low	Low	Low	Serious	Low	Moderate	Serious
Scharf et al. 2020 [59].	Moderate	Critical	NA	Critical	Low	Low	Moderate	Critical

Abbreviations: NI–Not enough information reported. NA–Not Applicable.

**Table 3 antibiotics-11-01311-t003:** Pharmacological targets of RCTs and cohort studies.

Pharmacological Target	References
100% *f*T>4-10xMIC	[40]
40% *f*T>4-6xMIC	[43]
60% *f*T>MIC	[44]
100% *f*T>1-10xMIC	[45,52]
100% *f*T>MIC	[46,47,51,57,59]
Css:MIC ≥ 1	[48]
Css:MIC ≥ 4	[48]
Css:MIC = 4–6	[49]
Css:MIC = 4–8	[50]
100% *f*T>4-5xMIC	[53,58]
100% *f*T>4xMIC	[55,57,59]
Specific carbapenem concentration	[39]
At clinician’s discretion guided by TDM result	[54,56]

**Table 4 antibiotics-11-01311-t004:** Effect of carbapenem TDM in health outcomes.

Study	Population	Improved Target Attainment?	Bacterial Persistence	Mortality	In-Hospital Stay	Length of Stay on ICU	Acute Kidney Injury	Toxicity or Adverse Effects	Treatment Efficacy	AMR
Randomised controlled trials										
Fournier et al. 2018. [39]	Burns patients (n = 17)	Not significant	*-*	-	-	-	-	-	No significant difference	-
De Waele et al. 2013. [40]	Non-renally impaired patients (n = 13)	Significant	No significant difference	No significant difference	-	-	-	-	No significant difference	-
Non-randomised comparator studies										
Machado et al. 2017. [44]	Burns patients (n = 16)	Not specified	-	No significant difference	-	-	-	-	No significant difference	-
Fournier et al., 2015 [46]	Critically ill burns patients (n = 109)	Variable success	-	No significant difference	TDM group significantly longer	-	-	-	No significant difference	-
Aldaz et al., 2021. [58]	Critically ill patients (n = 154)	Not specified	No significant difference	No significant difference	TDM group significantly shorter	No significant difference	-	No significant difference	Significant normalisation of infection biomarkers	-

Abbreviations: -, Not assessed; ICU, Intensive care unit; TDM, Therapeutic drug monitoring.

## Data Availability

Not applicable.

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
