# Peer review of "A Systematic Review of the Effect of Therapeutic Drug Monitoring on Patient Health Outcomes during Treatment with Carbapenems"

_antibiotics, 2022, doi:10.3390/antibiotics11101311_

Round 1

Reviewer 1 Report

Overall: The present study is a systematic review evaluating the effect of therapeutic drug monitoring of carbapenems on patient outcomes. Among 38 studies, only a qualitative assessment could be made as the studies were too heterogenous for meta-analysis. The evaluation reflected a number of studies suggestive of a correlation between TDM related target attainment achievement and clinical success.

Overall, these data are informative and interesting. I applaud the efforts of the authors in their work. Would recommend a couple minor copyedits before publishing.

Abstract

Lines 28-31: Run on sentence here is cumbersome. Would consider revising.

Results

Line 78: please fix ‘Error! Reference source not found’.

Author Response

  1. Line 78 Reference source not found.

Corrected to Table 1

  1. Line 79 Add a comma after study.

Comma added

Reviewer 2 Report

See attachment.

Author Response

According to instructions for authors: “in the text, reference numbers should be placed in square brackets [ ], and placed before the punctuation” ; in the article, you placed the references after the punctuation. Please, revise.

All punctuation moved after references.

- Line 69-70: remove the line break.

Amended

- Line 78: replace “Error! Reference source not found.” By “Table 1”.

Corrected to Table 1

- Figure 1: why “*” is present in “Records identified from*:”. There is no footnote.

Removed “*”

- Lines 93-106: you can indicate only once the reference [20] in the paragraph, not four times (lines 95, 97, 100 and 106). The next paragraph (lines 109-122) is correct: reference [21] is cited once.

Removed the reference in lines 97, 100, and 106. The reference in line 95 remains.

- Line 109: “et al.” is not in italics like in the rest of the article.

Et al has been italicised

 - Line 117: you did not define the SOFA score.

SOFA score defined

- Lines 117, 135, 152 and 269: the abbreviations “ICU”, “CRRT”, MIC”, “EUCAST”, ‘ARC” and “GFR” need to be defined the first time they appear in the text.

Defined

- Line 120: you indicated that “the number of patients was small”, can you precise the sample size in the study of De Waele et al.?

Added “(n=13, receiving meropenem)”

- Table 2, column “Study”: can you use the same presentation than in Table 1 (i.e., without abbreviations of the first names).

Removed first names

- Table 2: can you add a footnote for “NA” and “NI”?

Abbreviations moved from table title to footnote

- Table 2: the table was broken in two parts (on two pages). Can you merge the table on a single page?

Now on single page

- Table 3: what means “NRIS”? This abbreviation was not used in the text.

Replaced with cohort studies for consistency

- Line 196: why a page break after this sentence?

Page break is needed to orient the next page horizontally. Text has been moved so there is not a large gap of empty space.

- Line 220: remove the comment.

Comment removed

- Lines 299-300, you wrote: “A hand search was also carried out of references of relevant reviews”; but, in Figure 1, no article seems come from this hand search. Did you do this?

This was done but no additional articles were found, after de-duplication. Added “After de-duplication, no additional studies were found from the hand-search of relevant reviews” to line 69-70

- Line 310-311, you wrote: “non-randomised trials (cohort studies and quasi-experimental studies)”. Cohort studies and quasi-experimental studies are not trials.

Corrected by removing brackets

- Lines 310-312: you can add “clinical cases” in the type of study included.

Added

- Lines 340-345: can you add references which present each assessment tool used (RoB2, ROBINS- 1, OHAT)?

Included references

- The conclusions section was not totally supported by the results section. Some sentences of the conclusions could be moved in the discussion section.

Conclusion rephrased and added: “further studies are needed to investigate this hypothesis” to statement referring to TDM being a tool to suppress AMR.

- Regarding supplementary data: why “1946 to February 7, 2022” for MEDLINE search, and “1947 to 2022 February 7” for Embase search, and “1900-present” or “1975-present” for WoS search? In your materials and methods section, you wrote: “The search was conducted from database inception to 26 June 2020 and rerun on the 07 Feb. 2022”. The start date of inclusion in not the 26 June 2020?

The search was carried out twice. The first time was from database inception – 26 June 2020, which varied between the different databases. The search was then re-run on the 07 Feb 2022 to pick up additional papers that were published between 26 June 2022 – 07 Feb 2022. This was to follow the recommended guidance from the Cochrane Collaboration. To avoid future confusion the statement has been adjusted to “The search was conducted from database inception to 07 Feb. 2022”

- Regarding supplementary tables (S1, S2 and S3): for a better understanding and for improve the readability, can you use the same reference numbers (in square brackets) than in the article? Indeed, when we read a sentence in the article which indicated a reference number, we cannot directly consult the supplementary tables (we need to consult the references section in the article to see article reference before search the details of this article in the supplementary tables). Then, remove the references section in the supplementary data.

All reference numbers of supplementary material now consistent with article numbering

Reviewer 3 Report

The authors have done a systematic review of therapeutic drug monitoring on patient health outcomes during treatment with carbapenems.  Overall, the paper is well written and has good analysis.

Two minor corrections:

1.  Line 78 Reference source not found.

2. Line 79 Add a comma after study.

Author Response

The authors have done a systematic review of therapeutic drug monitoring on patient health outcomes during treatment with carbapenems.  Overall, the paper is well written and has good analysis.

Two minor corrections:

  1. Line 78 Reference source not found.

Corrected to Table 1

  1. Line 79 Add a comma after study.

Comma added

Reviewer 4 Report

The systematic review aimed to describe the effect of carbapenem TDM on health outcomes. The study was standardised conducted, however, several issues needed to be addressed.

 Major:

 1. Introcution & Discussion: It is known that AMR was a public concern and TDM was one of the practice to better promote appropriate use of drugs. However, compare to other drugs with narrow therapytic window that need TDM, the author need to sepecify more about the nessesity of why to conduct this review instead of stating that TDM of other antibiotics were reviewed. For instance, is carbapenem TDM was a guideline supported practice? Is carbapenem TDM was a prevelent service conducted? When and why did carbapenem TDM was needed? What types of patients would need carbapenem TDM? Is it for safty concern or effecacy concern, how did this review contribute to the clinical practice, etc. These questions are directly related to the rationale, quality and implication of this review. The introduction part need to illustrate more on the topic. The Discussion part also need to addess the mentioned concerns except for issues related to the quality of TDM itseft.

 2. Methods: The authored stated that “the references were screened by one reviewer, 10% of the references were randomly selected using a random number generator, for screening by a second reviewer to ensure consistency.” Please specify why did not let two reviewers concuted all the screening back to back independently? How did the author control the selection bias for the other 90% of the references?

Minor:

 1. The review searching period was no less than two years (Jun. 2020 to Feb. 2022). Please sepecify why choose such a time period instead of a more frequently used time period, for instance, 5 years?

 2.Figure 1 need to be described more explicitly. Why did the 731 records being excluded?

  3. There was some error in the article (line 78).

Author Response

Major:

  1. Introcution & Discussion: It is known that AMR was a public concern and TDM was one of the practice to better promote appropriate use of drugs. However, compare to other drugs with narrow therapytic window that need TDM, the author need to sepecify more about the nessesity of why to conduct this review instead of stating that TDM of other antibiotics were reviewed. For instance, is carbapenem TDM was a guideline supported practice? Is carbapenem TDM was a prevelent service conducted? When and why did carbapenem TDM was needed? What types of patients would need carbapenem TDM? Is it for safty concern or effecacy concern, how did this review contribute to the clinical practice, etc. These questions are directly related to the rationale, quality and implication of this review. The introduction part need to illustrate more on the topic. The Discussion part also need to addess the mentioned concerns except for issues related to the quality of TDM itseft.

Intro:

Added further context of this systematic review lines 43-64 and 80-81.

“Pharmacokinetic/pharmacodynamic (PK/PD) indices can help describe the relationship between the dose, concentration, and outcome[5]. Thus, where it is known that certain concentrations result in improved clinical outcomes, the PK/PD indices can be used as concentration targets to adjust doses towards. PK/PD indices also reflect different antibiotic action: time-dependent antibiotics (beta-lactams) are measured based on the amount of time unbound antibiotic is above the Minimum Inhibitory Concentration (MIC) (ƒT>MIC), where a fraction of the antibiotic is reversible bound to proteins; concentration-dependent antibiotics (aminoglycosides) are measured on the ratio of peak concentration to MIC (Cmax/MIC); and antibiotics that display time and concentration-dependent action are measured on the area under the curve to MIC (AUC/MIC)[6]. TDM is already carried out on some antibiotics such as aminoglycosides and glycopeptides[7]. Previous research has shown achieving Cmax/MIC targets [8-10], and AUC/MIC targets[11,12] in aminoglycoside treatment improves clinical outcomes. Extended high AUC exposures and high trough concentrations over a prolonged period have been associated with nephrotoxicity[5]. In glycopeptides, higher AUC/MIC ratios are associated with improved clinical outcomes [13-15]. Clinical studies of patients treated with beta-lactams have shown an association between clinical outcomes and beta-lactam exposure[16-25]. Because of the different PK/PD properties of different beta-lactams, it cannot be assumed that effectiveness of TDM of one beta-lactam will translate to all beta-lactams, so evaluations are needed by antibiotic class. Despite evidence of associations between clinical outcomes and PK/PD indices, few systematic reviews have assessed the impact antibiotic TDM has on clinical outcomes and impacts on AMR [26-28].”

Clinical studies have reported associations between PK/PD indices and clinical outcomes in carbapenems [16,17,19,24,25]

Discussion:

Further discussion added lines 245-269.

“Previous research has shown that beta-lactam TDM improves target attainment[33], and previous clinical studies have shown associations between outcomes and PK/PD indi-ces[16,17,19,24,25]. However, this review has identified that, whilst there is a previously observed association between outcomes and carbapenem target concentrations, there are no obvious improvements in outcomes when carbapenem TDM is implemented. This may be a result of study design, with inadequate power to detect a difference and/or how carbapenem TDM is carried out. To ensure that concentration targets are being met over the course of the treatment follow-up measurements must be taken. Just three of the in-cluded studies demonstrated improvements in target attainment[40,47,50], and frequency of concentration measurement was commonly not reported. A hurdle to successful TDM may be the accessibility of TDM services. The current gold-standard is HPLC-MS/MS, this requires specialised equipment, trained personnel, and time to prepare samples. More accessible TDM services, using a lab-based test that utilises commonly available equipment, such as an ELISA, or a ward-based point of care test, may allow for more frequent concentration measurements; to ensure concentrations are within target ranges over the course of treatment. This would be especially pertinent in patients with dynamic renal functions, or in patients receiving RRT [16]. This highlights the importance of the TDM clinical pathway, the steps required to carry out TDM: sampling, measurement, analysis of result, dosing regimen adjustment, follow-up measurements. Previous to this systematic review, TDM practice and the lack of standardisation in carbapenem TDM has not been discussed.

  1. Methods: The authored stated that “the references were screened by one reviewer, 10% of the references were randomly selected using a random number generator, for screening by a second reviewer to ensure consistency.” Please specify why did not let two reviewers concuted all the screening back to back independently? How did the author control the selection bias for the other 90% of the references?

This was done for pragmatic reasons, it wasn’t a fully funded SR which takes a team of people 10% of data extractions were independently checked by a second reviewer and there was a high degree of agreement between reviewers (of the 10% there were 0 differences when unblinded), negating the need for a full data extraction by 2 reviewers. As this has not been considered a problem by 3 out of the 4 reviewers, we feel our approach is reasonable. We would be willing to repeat data extraction for the remaining 90% of papers but we feel that this is highly unlikely to change the conclusions of the manuscript and will inevitably delay publication. 

Minor:

  1. The review searching period was no less than two years (Jun. 2020 to Feb. 2022). Please sepecify why choose such a time period instead of a more frequently used time period, for instance, 5 years?

The search was carried out twice. The first time was from database inception – 26 June 2020, the search was then carried out on the 07 Feb 2022 to pick up additional papers that were published between 26 June 2022 – 07 Feb 2022. To avoid future confusion the statement has been adjusted to: “The search was conducted from database inception to 07 Feb. 2022”

 2.Figure 1 need to be described more explicitly. Why did the 731 records being excluded?

This was the title and abstract screen to filter out obviously irrelevant studies and highlight potentially relevant studies for the full-text screen. It is not common to present the reasons why these studies were excluded, however they were excluded under the same criteria as the full-text screen, wrong antibiotic, not TDM, no clinical outcomes, wrong study type, not in English. The majority of exclusions at the title and abstract stage were from wrong antibiotic and wrong study type, as these are easily decerned from the title and abstract.

  1. There was some error in the article (line 78).

Corrected to Table 1

Round 2

Reviewer 4 Report

Thank you for the revision as required.